# Producing aerosol size distributions consistent with optical particle counters measurements using space-based measurements of aerosol extinction coefficient

Nicholas Ernest [1], Larry W. Thomason [2], and Terry Deshler [3,4]

[1]ADNET Systems, Inc., Hampton, VA 23662, USA
[2]NASA Langley Research Center, Hampton, Virginia, 23681, USA (retired)
[3]Department of Atmospheric Science, University of Wyoming, Laramie, Wyoming, 82070, USA
[4]Now at Laboratory for Atmospheric and Space Physics, University of Colorado, Boulder, Colorado, 80303, USA

**Correspondence:** Larry W. Thomason (l.w.thomason@scienceronin.org)

**Abstract.** Stratospheric aerosol has been observed by several long-lived observational systems. These include the University of Wyoming series of balloon-borne optical particle counters (OPCs) (1971-2020) and the Stratospheric Aerosol and Gas Experiment (SAGE) series of instruments and particularly SAGE II (1984-2005). Inferences of aerosol surface area density (SAD) and volume density are straightforward using data from OPCs. Conversely, many numerical methods to infer size distributions and SAD have been applied to SAGE II observations but all are limited by the restricted number of independent wavelengths of the SAGE optical measurements. We have developed a new method that uses OPC observations to constrain SAGE II inferences of aerosol properties. We start by noting that whatever the details of the underlying size distribution, the SAGE II measured aerosol extinction coefficient ratio (525 to 1020 nm) must reflect the shape of the underlying aerosol size distribution for particles that dominate the extinction coefficient values (roughly radii from 0.1 to 0.5 μm). Since this extinction ratio can be easily calculated from OPC measurements, we use the OPC size distribution measurements, across a broad range of aerosol levels from background to highly volcanic, to compute the associated 525 to 1020 nm extinction coefficient ratios for each measurement. We then sort the OPC measurements by these ratios (across a range of roughly 1 to 6) into discrete ratio bins and derive mean bimodal log-normal size distributions for each bin using a particle swarm optimization. These fits can be applied to SAGE II observations without the need for further retrieval calculations effectively producing an OPC-like product consisting of the six bimodal parameters for all SAGE II observations. This method successfully captures the median behavior of the OPC inferences of bulk parameters like aerosol surface area and volume density, although we also observe a significant altitude dependence particularly in the lower stratosphere. In addition, there are occasional deviations of SAD from the fit behavior by as large as a factor of 10 for individual OPC measurements of SAD, primarily due to variations in small radii particle number density (roughly those smaller than about 0.15 μm). The presence of such particles is effectively invisible to extinction coefficient measurements such as those by SAGE II.

# 1 Introduction

Space-based measurements of stratospheric aerosol extinction coefficient have been made continuously since the 1978 launch of the Stratospheric Aerosol Measurement (SAM II) instrument aboard the Nimbus-7 spacecraft. For practically the same period of time these measurements have been used to infer underlying properties of the aerosol focused on the aerosol size distribution (ASD) and properties that impact chemistry and climate such as aerosol surface area density (SAD) and total aerosol mass or volume density (VD). Among the earliest efforts to infer ASD using space-based measurements was use of the 450 and 1000 nm aerosol extinction coefficient measurements by the original Stratospheric Aerosol and Gas Experiment instrument (SAGE, 1979-1981) to fit a single mode log-normal of fixed width (1.6) and inferring the mode radius and number density (Yue et al, 1983). Since the stratospheric aerosol size distribution tends roughly to resemble a single mode log-normal (SLN) (Pinnick et al., 1976), though other mathematical forms exist, the SLN remains a common starting point for many ASD algorithms based on space-based observations of aerosol extinction coefficient (e.g., Russell et al., 1996, Arfeuille et al., 2013, Nyaku et al., 2020, Knepp et al. 2023) often making use of the long-lived SAGE II mission (1984-2005). SAGE II measurements, with aerosol extinction coefficient measurements at 4 wavelengths (385, 452, 525, and 1020 nm), remain an object of considerable scientific attention given that they include observations of part of the recovery from the 1982 eruption of El Chichón, the 1991 eruption of Mt. Pinatubo and its recovery, and a relatively volcanically quiet, low aerosol loading period from 2000 to the end of its mission in 2005. As a result, this record remains a core part of the long-term stratospheric aerosol record (Kovilakam et al., 2023) and still plays a significant role in the study of the impact of volcanic activity on climate and the processes that lead to ozone destruction (Rieger et al., 2020; Revell et al., 2017; Pauling et al., 2023).

After 40 years, it might be expected that there would be a generally accepted approach (or approaches) from which robust determinations of ASD from the SAGE II measurements are routinely inferred and this topic would be of limited further effort, however this is not the case. One reason for the proliferation of diverse methods for solving for aerosol properties is that, while the retrieval methods are almost always mathematically straightforward, all retrieval methods effectively constrain the ASD solution space such that not all mathematically (as opposed to physically) plausible solutions for ASD are allowed. This is necessary because the measurements contain insufficient information to uniquely identify the ASD across the span of radii that are relevant to chemistry and climate. For instance, Thomason et al. (2008) showed that SAGE II aerosol extinction measurements have an explicit minimum SAD that is consistent with observations, but an upper limit that is unbound (effectively infinity) by allowing very large numbers of very small and very ineffective scattering particles. Retrieval algorithm constraints often take the form of a fixed mathematical form for the aerosol size distribution (e.g., the SLN) or constrains the way aerosol number density varies as a function of particle radius (e.g., smoothness). Most a priori constraints are not on face value unreasonable but, when applied, fundamentally affect the outcomes for ASD and bulk property retrievals in ways that are not easy to account for in an uncertainty estimate and can vary substantially from constraint to constraint. Unsurprisingly these factors point out the lack of robustness in SAGE II-based ASD and other aerosol property retrievals for which no solution based solely on the measurements is possible.

Given the limited number of SAGE II measurement wavelengths and the correlation between particularly the short wavelength channels (385, 452, 525 nm), inferring an ASD more complex than a SLN (with 3 free parameters) such as a bimodal size distribution (with six) is impossible. However, in situ measurements of stratospheric aerosol size distributions are often more complex than a SLN. For instance, the University of Wyoming optical particle counter (WOPC) measurements are often fit best with a bimodal log-normal aerosol size distribution (Deshler et al., 2003, 2019). Both modes do not necessarily contribute significantly to a computed aerosol extinction coefficient at SAGE II wavelengths. In many cases the computed extinction at SAGE II wavelengths does not differ significantly between a fitted SLN distribution and a bimodal distribution. Both modes, however, are often important in the estimation of aerosol bulk properties like SAD, which can be dependent on small particles, primarily missed by optical extinction measurements, (Deshler et al., 2003). While we will make extensive use of WOPC data to infer aerosol bulk properties from SAGE II retrievals, this is not primarily a validation or intercomparison of the measurements of these two instruments which have been presented elsewhere (Hervig and Deshler, 2002; Deshler et al., 2003; Kovilakam and Deshler, 2015, Deshler et al., 2019). To a greater degree, we are trying to determine whether it is possible to infer the magnitudes and variability observed in WOPC-derived key parameters like SAD from SAGE II measurements. In that regard, we are treating the WOPC measurements as a test bed for SAGE II retrievals.

Therefore, in this paper, we discuss the SAGE II measurements and demonstrate some of the limiting factors for ASD inferences. We then estimate aerosol extinction coefficient at SAGE II wavelengths and SAD using the WOPC data alone and show how WOPC SAD varies with computed aerosol extinction and its wavelength dependence. The degree to which this relationship is well-behaved directly addresses how well the SAGE II measurements can be used to infer ASD or SAD consistent with WOPC values. We demonstrate that, while the median behavior of WOPC observations can be replicated, there remain substantial SAD and VD positive outliers, primarily in the lower stratosphere, that are larger than the median value by factors as large as 10. While it applies only to the median behavior, we produce a WOPC-based bimodal log-normal ASD, varying with aerosol extinction coefficient wavelength dependence, that potentially allows a bimodal aerosol size distribution to be assigned to any SAGE II multi-wavelength stratospheric aerosol extinction coefficient measurement set. While the analysis and its outcomes are strictly only relevant to WOPC/SAGE II comparisons in the mid-latitudes of the Northern Hemisphere, the outcomes reflect a fundamental limitation on what is possible for aerosol property estimates from SAGE-like measurements. Work is ongoing now to extend this analysis to SAGE III/ISS.

## 2    Some issues related to estimating size distribution using SAGE II data

The SAGE II aerosol extinction coefficient ensemble consists of measurements at 4 wavelengths (385, 452, 525, and 1020 nm) that usually extend from the upper troposphere to 40 km. Assuming Mie scattering, the aerosol extinction coefficient, $k_\lambda$, can be mathematically expressed as

$$k_\lambda = \int\limits_0^\infty \pi r^2 Q_\lambda(r, m_\lambda) \frac{dn(r)}{dr} dr \qquad (1)$$

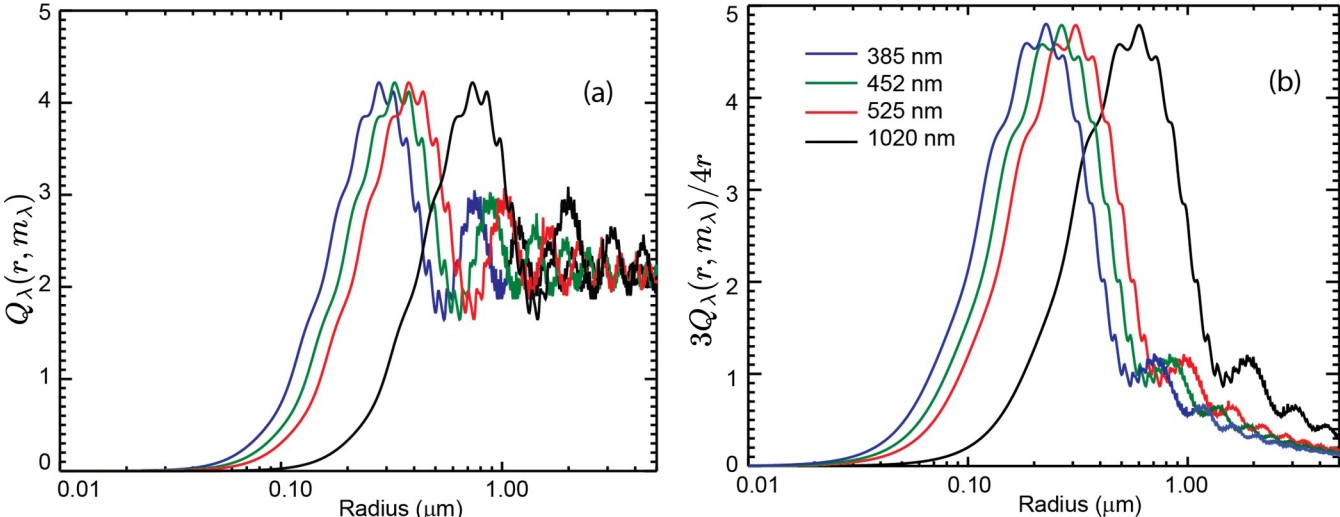

**Figure 1.** (a) Mie extinction kernels and (b) kernels scaled to per unit aerosol volume times wavelength, for the SAGE II channels assuming spherical water/sulfuric acid droplets at 220 K and a composition of 75% H2S04 and 25% H2O. The real refractive indices used were 1.432, 1.432, and 1.421 for 453, 525, and 1020 nm respectively with zero for all imaginary parts.

or

$$k_\lambda = \int_0^\infty \frac{3}{4r} Q_\lambda(r, m_\lambda) \frac{dV(r)}{dr} dr \qquad (2)$$

where $Q_\lambda$ is the Mie kernel for particles of radius $r$ and with an index of refraction $m_\lambda$, and the aerosol size distribution is $dn(r)/dr$ in number per unit radius and $dV(r)/dr$ in volume of aerosol per unit radius. Figure 1a shows the values of $Q_\lambda(r,m_\lambda)$ using a refractive index typical of stratospheric conditions and sulfuric acid-water aerosol. Figure 1b shows the

90    per unit volume kernels weighted by the measurement wavelength or $3Q_\lambda(r,m_\lambda)/4r$. From these figures, it is clear that the SAGE II measurement ensemble does not contain significant information for particles much less than 0.1 μm even for the shortest wavelength measurement. As a result, estimates of total number density and similar parameters dependent on low-order moments of the aerosol size distribution (e.g., number density) are not well constrained by the measurements and, in fact, depend on how a retrieval process fills this information gap. Thus, while SAGE II measurements can almost always be used

to find a unique log-normal (or similar low free parameter) aerosol size distribution that reproduces the measurements, there is no guarantee that all high value geophysical parameters like SAD will be adequately calculated.

There are further complications in performing SAGE II ASD retrievals. The measurements at 385 nm are not considered reliable except at relatively high extinction coefficient values ($> 10^{-3}\,\mathrm{km}^{-1}$) (Thomason et al., 2018) and are not recommended for general use, reducing available measurements for ASD retrievals to only 3. In addition, as Figure 1b shows, the 2 remaining

short wavelength measurements (452 and 525 nm) have significant overlap in their extinction kernels and thus provide limited unique information between them, particularly in light of their associated uncertainties. This can be demonstrated with a

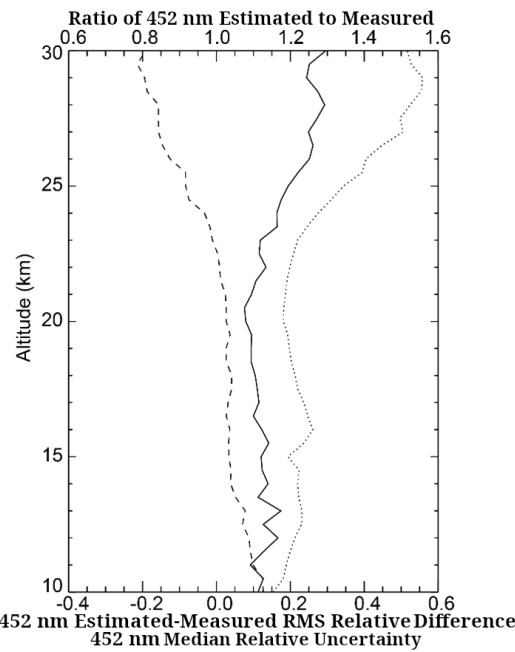

**Figure 2.** For the Northern Hemisphere (>20° N) in April 1999 at 452 nm: 1) ratio of estimated to measured aerosol extinction coefficient (dashed, top scale), 2) relative RMS difference between estimated and measured aerosol extinction coefficient (solid, bottom scale). 3) median relative aerosol extinction coefficient measurement uncertainty (dotted, bottom scale).

relatively simple exercise. First, we compute an Angstrom coefficient using SAGE II extinction coefficient data at 525 and 1020 nm and then use this value to extrapolate to aerosol extinction coefficient at 452 nm using one of the former measurements as the base. While the mean difference between the estimated and measured aerosol extinction coefficient is primarily a measure

of how well the extrapolation works, its relative RMS (root mean square) mean difference is a measure of how much unique information exists in the 452 nm measurement. Crudely, if the relative RMS mean difference is greater than the 452 nm measurement uncertainty, then it is possible that there is some usable additional information contained in the measurement.

In Figure 2, we show the outcome for April 1999 with measurement locations north of 20° N where observations at all 3 wavelengths exist and where extinction coefficient is less than 0.01 $\text{km}^{-1}$ (a crude cloud filter) and greater than $10^{-5}$

$\text{km}^{-1}$, below which measurement quality decreases rapidly. In this figure, we show the mean ratio between the predicted extinction coefficient and the measured extinction coefficient at 452 nm as a function of altitude (dashed line, scale at the top). On average, the Angstrom extrapolation does well between 13 and 24 km where the predicted value is within 5% of the observed values. The departures increase to about +10% at 10 km and -20% at 30 km, primarily demonstrating the limitations in the interpolation method. This figure also shows that the relative RMS mean difference (solid line, bottom scale) between

estimated and measured aerosol extinction coefficient, a stand-in for inferred noise, is routinely about half the size of the reported measurement uncertainty (dotted line, bottom scale). In this context, the differences between reported and inferred uncertainties are likely somewhat exaggerated due to the correlation in measurement uncertainty particularly among SAGE

II's short wavelength channels and, to a lesser extent, that the reported uncertainties contain both systematic and precision elements (Damadeo et al, 2013). Nonetheless, the degree to which the 452 nm channel can be inferred from the values at 525 and 1020 nm strongly suggests that either there is limited variability in aerosol size distribution for particles which control the 525 to 1020 nm extinction coefficient ratio or that, if significant variability does exist, then the ability of the 452 nm channel to illuminate that variability is very low.

If the ability of the 452 nm channel to illuminate variability in the ASD is low, as we will show below, then fitting a meaningful low parameter size distribution, like a SLN, is problematic. This assessment is corroborated by past efforts to infer size distributions from these measurements in which single mode log-normal fits with SAGE II data produce distributions that are rather narrow (e.g., Wang et al., 1989), and findings that it is possible to fit the extinction coefficient measurement spectra using a vanishingly narrow distribution (a delta function) (Thomason et al. 2008). We conclude that the SAGE II aerosol extinction measurement ensemble has at best two pieces of information that are most clearly represented in the overall magnitude of extinction at 525 and 1020 nm and their extinction coefficient ratio (or the extinction coefficient spectral slope). This is in basic agreement with assessments of the information content of the measurements (e.g., Thomason et al., 1997). As well a similar conclusion was reached by Thomason and Poole (1993) using a different technique. With so little information contained in the measurements, essentially all SAGE II aerosol size distribution retrievals have little recourse except to be dependent on the retrieval method. In other words, the outcomes from the retrieval process are likely controlled by the assumptions made at the outset of the effort and the robustness of the inferences are debatable.

While we are focused on SAGE II, it is worth considering whether other measurement types such as the limb-scatter technique employed by OSIRIS can be used to infer aerosol size distributions in a similar way. It is clear from Figure 1, that adding short wavelength measurements (e.g., at 385 nm) would increase the information about the small particles present. Practically, however, robust measurements of stratospheric aerosol extinction coefficient at wavelengths much shorter than the 385 nm channel on SAGE II are difficult due to the effects of molecular scattering and absorption by ozone and other gases. Simply increasing the number of visible and near-infrared measurements may, through repetitive information, improve the resolution of size distribution retrievals in the 0.1 to 0.5 μm radius range; however, the degree to which this is true depends on the precision of the measurements and the details of the measurements' spectral location. Still, such an instrument (e.g., the current SAGE III/ISS instrument) may not radically improve the ability to infer aerosol size distributions which could represent the small particles (<0.1 μm radius). A possible long-term solution would be to add measurements where sulfuric acid aerosol strongly absorbs in the infrared. The sulfuric acid extinction kernels in the infrared are relatively flat across radii relevant to stratospheric aerosol and are, thus, a near, but not exact, measure of the total volume of aerosol present in the measurement volume (Thomason, 2012). Combined with visible and near-infrared measurements (where scattering dominates) infrared measurements could provide some constraint to what occurs at smaller particle sizes (Thomason, 2012; Boone et al., 2023), though most likely this would still require significant constraints to the possible solutions. Volume is inherently insensitive to small particles, in contrast to surface area.

A characteristic of almost all size distribution retrievals is that they tend to stand on their own apart from information arising from in situ measurements of aerosol size distributions (e.g., by optical particle counters) beyond very general considerations

such as that the aerosol size distribution is generally compatible with single or multimodal log-normal aerosol size distributions. Given the weak information content of the SAGE II measurements, we now consider the possibility of using in situ information much more explicitly. In the following sections, we will examine the ability to use the in situ measurements from the WOPC to assess the variability in size distributions as a function of the 525 to 1020 nm extinction coefficient ratio and attempt to infer 'WOPC-compatible' aerosol size distributions from SAGE II extinction coefficient measurements. We will evaluate the success of this effort primarily by how well such inferences can reproduce WOPC-like SAD values.

## 3   University of Wyoming OPC measurements

The University of Wyoming (UW) in-situ balloon-borne measurements of aerosol size distributions have been made continuously since 1971 (Deshler et al., 2003). Vertical profiles of size resolved cumulative aerosol concentration are provided along with unimodal/bimodal log-normal fits. The number density profiles are provided at full resolution and 0.5 km resolution, the size distribution fits at 0.5 km resolution, and these include calculated SAD and VD. The instrument originally used was developed by Rosen (1964) and utilizes the method of dark field microscopy, focusing diffracted light from a particular angle onto a photomultiplier tube, which converts photons to voltages. The fundamental measurement of an OPC is the scattered intensity, or voltage, from an illuminated particle. Calibrations and the OPC counter response function then associate these voltages with a particle size, and the number of particles above a certain size is accumulated into size bins. Light scattered by aerosol particles was originally measured at a 25° forward angle in the UW project, the Dust instrument measuring 2-4 sizes from 0.15 - 0.3 μm. All sizes are given as radius. This was changed to a 40° angle in 1991 to allow for size resolution between 0.3 and 10.0 μm (Hofmann and Deshler, 1991; Deshler et al., 2003), the WOPC. The switch to a laser particle counter began in 2008, measuring side scatter in a large solid angle centered on 90° (Ward et al., 2014), the WLPC. The WOPC provided 8-12 sizes, 0.19 – 2.0/10.0, while the WLPC provided 8 sizes, 0.09 – 4.5μm. The measurements are made from the surface to ∼30 km. Included with most measurements is a second instrument to measure all particles > 0.01 μm using a condensation nuclei counter which measures particles by growing the particles to optical detection by supersaturating the air stream with ethylene glycol vapor (Rosen and Hofmann, 1977; Campbell and Deshler, 2014). The data from the Dust counter from 1971 – 1988 are available at https://ndacc.larc.nasa.gov/. For flights after 1988, with the Dust, and for the WOPC (1989-2013), and WLPC (2008-2020) data see Deshler (2023). This data record is now being extended with a new OPC, the LOPC, from Boulder, Colorado (Kalnajs and Deshler, 2022), with > 50 channels, 0.15 – 10 μm. These data are also available from Deshler (2023). The instrument with the most overlap with SAGE II is the WOPC and will be used as the reference OPC through the rest of this paper.

The size distribution measurements are fit with a unimodal or bimodal log-normal distribution, depending on the count of channels available and which shape produces the best fit. A log-normal size distribution consists of the total number concentration $N_j$, the median radius $\mu_j$, and the distribution width $\sigma_j$ for each mode $j$. The unimodal/bimodal log-normal size distribution is given by the following integrals where $a$ is the integration variable:

$$N(r > r_{ch}) = \int\limits_{r_{ch}}^{\infty} \left[ \sum_j \frac{dn_j}{dln(a)} \right] dln(a) = \int\limits_{r_{ch}}^{\infty} \left[ \sum_j \frac{N_j}{\sqrt{2\pi}ln(\sigma_j)} exp\left( \frac{-ln^2(a/\mu_j)}{2ln^2(\sigma_j)} \right) \right] dln(a) \qquad (3)$$

However, to better account for instrument counting efficiency, this equation has been modified to reflect the instrument's ability to count aerosols at the channel boundary (Deshler et al., 2019). The equation that is used to fit the measured concentrations is now:

$$N_{ch} = \int\limits_0^{\infty} \left[ \sum_j dn_j/dln(a) \right] \cdot CEF_{ch}(a) \cdot dln(a) \qquad (4)$$

Where CEF is the counting efficiency of the OPC instrument which is modelled as a cumulative Gaussian distribution:

$$CEF_{ch}(r) = \frac{1}{2} \left[ 1 + erf\left( \frac{r-\mu}{\sqrt{2}\sigma} \right) \right], \qquad (5)$$

where $erf()$ is the error function. In this equation $\mu$ is the size of the 50% counting efficiency point, and is the size reported in the WOPC data files. The other parameter, $\sigma$, is the rate at which the instrument counting efficiency approaches its limits of 0 and 1. The previous method of fitting assumed a perfect efficiency of the instrument to count all particles above the target radius and none below. Accounting for the realistic counting efficiency of the instrument (Deshler et al., 2019) has significantly improved the agreement between extinction coefficient computed using the resulting WOPC size distributions with those measured by SAGE II. Figure 3 shows an example of the OPC channel measurements during a period of high aerosol loading, and the fitted bimodal log-normal distribution for the data using the new fitting method.

## 4 Variability of SAD to extinction ratio as a function of extinction ratio

Using WOPC size distributions, it is straightforward to compute SAD and VD using analytic functions. It is also straightforward to compute aerosol extinction coefficient at any wavelength using these size distributions and Equation 1. For these calculations, we assume that aerosol is composed of sulfuric acid-water that, in turn, defines the refractive index used to compute the Mie kernels. This is usually a very appropriate assumption, but significant exceptions can occur particularly following injections of smoke into the stratosphere or ash and, to a lesser extent, by the presence of organic aerosol or other non-absorbing, non-sulfuric acid aerosol. While several large smoke events have occurred over the past decade, they are a relatively minor component of the SAGE II aerosol record (Thomason and Knepp, 2023) and will not be considered further in this discussion though for some specific instances and for other instruments including the on-going SAGE III/ISS mission, composition cannot be as easily ignored. Using only WOPC data, we compute three sets of ratios (the SAD to 1020 nm extinction coefficient ratio (SADR), the VD to 1020 nm extinction coefficient ratio (VDR) and the 525 to 1020 nm extinction coefficient ratio (R)) in two altitude

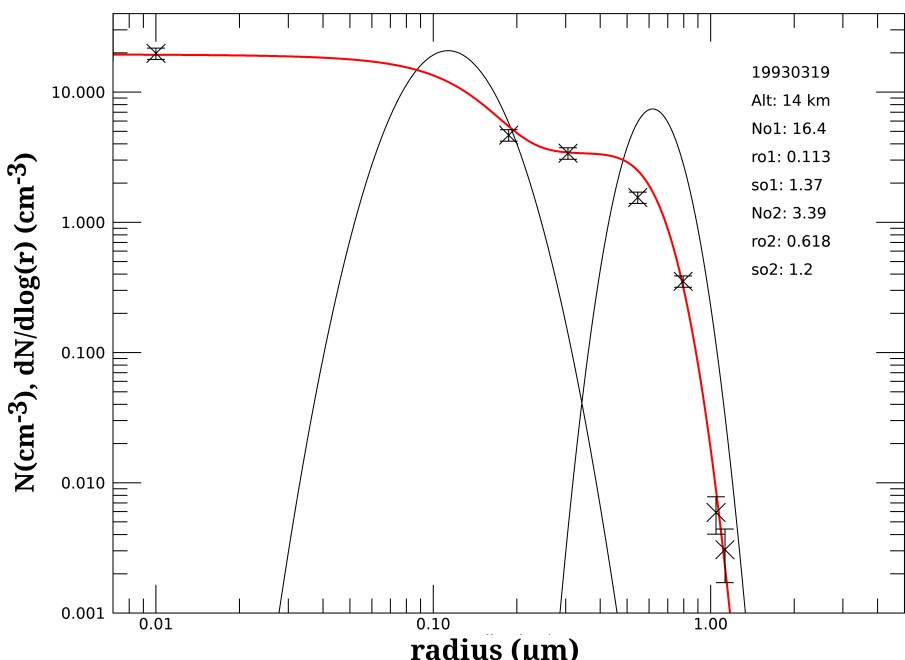

**Figure 3.** Example of WOPC measurements (black crosses). The black lines show differential size distributions for the two modes fit during the processing of WOPC data. The cumulative bimodal log-normal distribution is shown in red. Error bars are shown accounting for the Poisson error. The legend shows the date the measurements were made, the altitude (Alt), the total particle counts for the coarse and fine mode (No1/No2), the median radii (ro1/ro2), and the widths (so1/so2) in each mode.

ranges (13 to 19.0 and 19.5 to 25 km) for the period where WOPC overlaps with the SAGE II mission. This period spans the heavily volcanic Mt. Pinatubo period as well as the fairly quiescent period between 1999 and 2005.

Figures 4a and 4b shows the comparisons of SADR versus R for the two altitudes ranges. Organizing these plots by extinction ratio makes sense as the underlying size distribution, in the tail of the size distribution that dominates the extinction calculation, must be similar to all others that produce a roughly similar R. Superficially, the distribution of data on these curves are similar though it is clear that the lower altitude range shows much more scatter than the higher altitude set. The medians of SADR and VDR for both altitudes are the same for R near 1 but diverge and maintain a difference of almost a factor of 2 for much of the range of R values.

The values of SADR show a very non-linear conversion between SADR and R which varies from about 1500 for R ~1 to ~50000 for R around 6. While there are differences in the details, the actual conversion factors for extinction coefficient to SAD are not wildly different than those from Thomason et al. (2008). The distribution of scatter around the median SAD line is clearly not Gaussian and shows a significant positive (upward in the figures) skewness. The scatter and skewness of these points demonstrate how difficult it is to infer SAD from SAGE II measurements even with an assist from in situ observations

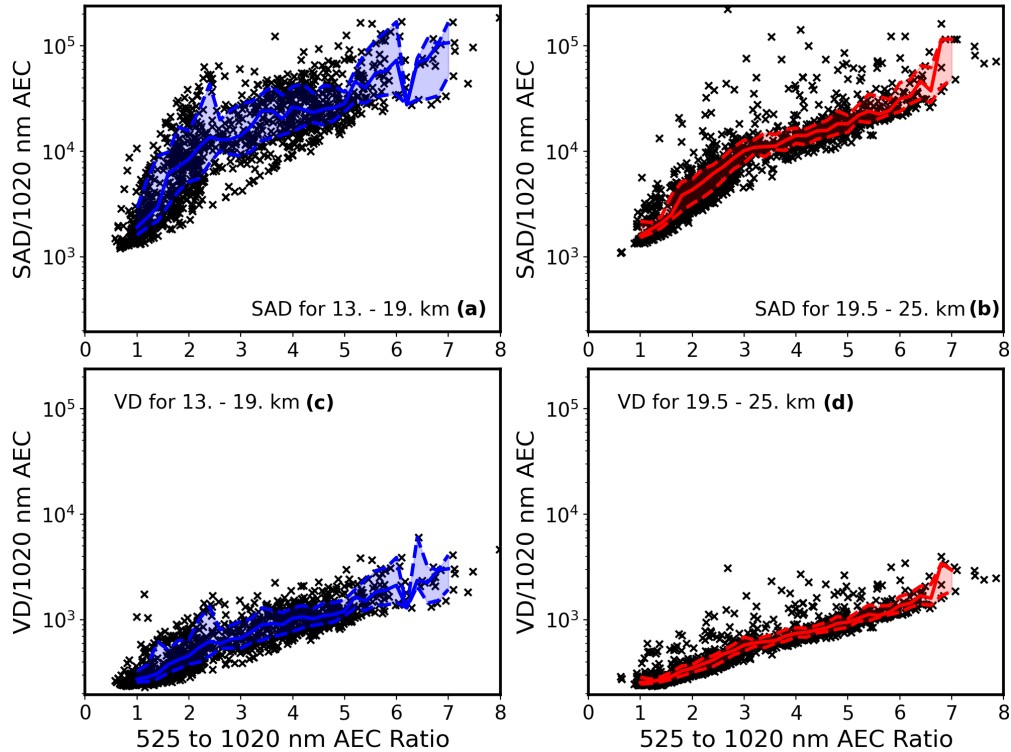

**Figure 4.** (a) and (b) show SAD/1020 nm aerosol extinction coefficient (AEC) to 525/1020 nm aerosol extinction ratio for the low and high altitude group respectively. The dashed lines show the 20th and 80th percentile values in bins of 0.2 ratio. The solid line shows the median values. (c) and (d) are the same as (a) and (b) but show VD/1020 nm extinction coefficient to 525/1020 nm extinction ratio.

like the WOPC. The extremes in SADR ranges are well over an order of magnitude at some values of R. At a more restricted scale, the range of the 20th and 80th percentile levels of SADR is around a factor 4 for R between 1 and 2.5 for the lower altitude range and decreases to around a factor of 1.7 for R around 5 in the lower altitude analysis. In the higher altitude analysis, the 20th and 80th percentile SADR range is notably narrower and is between 1.5 to 2 factor for values of R between 1 and 6. These ranges are larger, particularly, at low values of R, than those estimated in previous analyses (e.g., Thomason et al., 2008). The range in SADR at any given value of R is almost exclusively due to variations in small particle number concentrations that are poor scatterers and thus not reflected in any SAGE II-like measurement. Thus, while it may be possible to reproduce some of the WOPC SAD behavior from SAGE-like measurements, it is clearly impossible to reproduce all of the SAD variability observed by the WOPC.

The analysis of WOPC VDR versus R is better behaved as VD is more dependent on large particles than the lower order moment of the size distribution, SAD. In this case, the maximum range in both altitude ranges, shown in Figures 4c and 4d, are mostly less than 10 at the lower altitude range and, for the higher altitude range, less than 6. Similarly, the range between the 20th and 80th percentile curves is smaller for VDR than SADR. Around a factor of 2 difference for values of R less than

2.6 in the lower altitude analysis, and a factor 1.5 for R between 2.8 and 5. For the upper altitude analysis, the percentile range fluctuates between a factor of 1.1 and 1.4 for R until R exceeds 6.8 where the range increases.

## 5 Finding average size distributions

While the large outliers in SADR and VDR, particularly in the lower altitude range, cannot be captured using these extinc-
240 tion coefficient measurements, there is still some ability to capture median behavior and, therefore, there is some utility to associating SAGE II-like aerosol extinction measurements with WOPC compatible size distributions. Therefore, we pursue the development of representative bimodal log-normal size distributions for the WOPC as a function of the inferred 525 to 1020 nm aerosol extinction ratio. We used only WOPC measurements for which a bimodal log-normal distribution is most appropriate and exclude those for which only a unimodal size distribution is applicable. Data are analyzed in the two altitude
ranges used above, 13-19 km and 19.5-25 km, reflecting the observed differences in the distribution of inferred SAD/1020 nm ratio for those altitude ranges. We further subdivide the data by inferred 525 to 1020 nm extinction coefficient ratio into bins 0.2 width for extinction coefficient ratios from 1.0 to 8.2.

To simplify from 6 parameters, we scale all size distributions by the total particle number so that there are 5 free parameters to retrieve for bimodal distributions: the fraction of the data in the $1^{st}$ (small) mode, and the width and median radius of both the
250 $1^{st}$ and $2^{nd}$ modes. To retrieve these parameters, we employ a particle swarm optimization algorithm (Hu and Eberhart, 2002), where the many individual sets of WOPC data are referred to as particles in the algorithm's nomenclature, and the particles "fly" through parameter-space trying to minimize or maximize some function. This approach is unique in that it requires only the objective (or in our case minimization) function, and it is not dependent on gradients or derivatives of this function. This makes it fairly simple on the mathematical complexity scale of retrieval algorithms while, as we find, providing robust solutions. We
define the objective function, $OF$, to be the sum of errors for each of the unknown parameters or

$$OF = r_{01\_err} + s_{01\_err} + r_{02\_err} + s_{02\_err} + f_{err} + R_{err} \cdot w \tag{6}$$

where $r_{01\_err}/r_{02\_err}$ are the two median radii errors, $s_{01\_err}/s_{02\_err}$ are the errors for the widths of the modes, $f_{err}$ is the error for the ratio of the concentration of the first mode to the total concentration, $R_{err}$ is the extinction coefficient ratio error, and $w$ is a weight. The value of $w$ is selected to prioritize the target extinction ratio (the bin center) among possible solutions
since we value this outcome for this exercise. The error values are all positive as they are based on the absolute value of the difference between the particle's parameter value and the parameter median, divided by the parameter standard deviation, defined here:

$$"parameter\ error" = |("particle\ parameter\ value"\ - "parameter\ median")|/"parameter\ standard\ deviation" \tag{7}$$

where the median and standard deviation values for a parameter are determined from all the values within a particular
extinction ratio bin, and the "particle parameter value" is a particle's value for that particular parameter in the parameter space.

The extinction ratio error ($R_{err}$) is simply the current particle's calculated extinction ratio value minus the center of the target extinction ratio bin. In this method, many individual sets of WOPC data are effectively used to explore the parameter space through a series of iterations, $t$, to find a global minimum in $OF$ for the collection of particles in each target extinction ratio bin. The iterative process is given by

$$OF[t+1] = OF[t] + v[t] \tag{8}$$

where $v[t]$ is a 'velocity' parameter governed by an attraction to an individual particle's best value and the best value found among all of the particles in the bin. Where the best value is defined as the position in parameter space which results in the greatest minimization of the objective function. 'Velocity' is given by

$$v[t+1] = bv[t] + dv[t] \tag{9}$$

where

$$dv[t] = w_1 c[p_{best}(t) - p(t)] + w_2 c[g_{best}(t) - p(t)] \tag{10}$$

$b$ is a velocity damping factor, $w_1$ and $w_2$ are weights, $c$ is a uniform random deviate, $p$ is the current parameter set, $p_{best}$ is the parameter set yielding the lowest value for $OF$ found for the particle among all previous iterations, $g_{best}$, or global best, is the parameter set within $p_{best}$ that yielded the minimum value of $OF$ for any particle at any iteration. The variable $b$
controls roughly how quickly the solution moves in its current direction while the random perturbation created by the use of $c$ influences how strongly the solution can 'change directions' or explore the solution space to reduce the $OF$ value for the best individual position and the best overall solution. The weights $w_1$ and $w_2$ effectively control whether the degree to which the solution search can explore the full space ($w_1$>$w_2$) or pushes more directly toward the current consensus best or 'swarm' solution ($w_2$>$w_1$). In swarm optimization, this is referred to as weighting between exploration versus exploitation. For this we
have chosen the weights to slightly emphasize the attraction of the particles toward the global best, prioritizing exploitation of the particles. The $v[0]$ values are random perturbations of the initial variable values which are driven toward more instructive values by subsequent iterations. Obviously, there are a number of empirical knobs to turn if a solution isn't found easily. This generally depends on the character of the data, its variability and noise. In practice, we found that varying the values of $b$, $w_1$ and $w_2$ does not strongly affect the ultimate solutions though sometimes how rapidly it approached them.

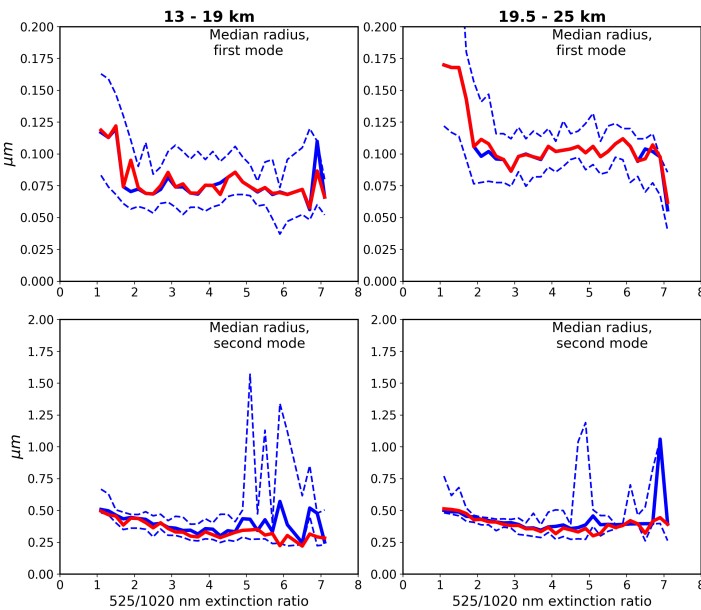

**Figure 5.** The two median radii parameters of a bimodal size distribution are shown for 13-19 (left) and 19.5-25 (right) km as a function of the 525/1020 nm extinction ratio. The median (bold blue) and 20[th] and 80[th] percentile (dashed blue) values are shown for each extinction ratio bin. Results of the swarm optimization fit are shown in red.

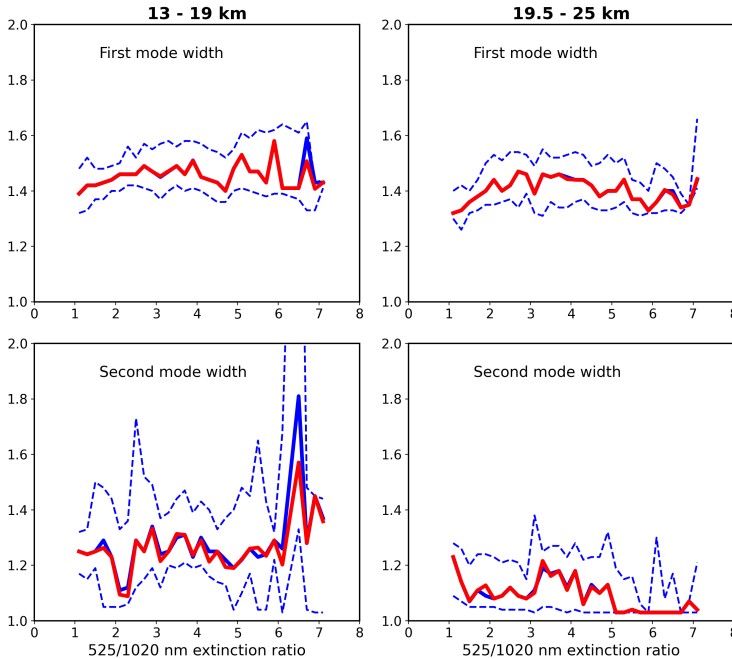

**Figure 6.** The same as Figure 5 but for the two mode widths.

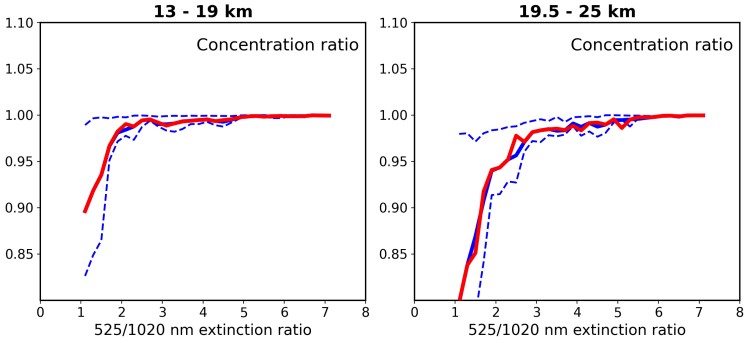

**Figure 7.** The same as Figure 5 and Figure 6 but for the concentration ratio.

Figure's 5, 6, and 7 show the final fit values for the lower and upper altitude groups for each of the 5 parameters. We also include the 20th and 80th percentile values for individual fits provided by the WOPC dataset and the median value (as a function of extinction ratio), comprised of 1541 WOPC values for the lower altitude group and 1515 values for the upper altitude group. We find, in general that the parameters found by the swarm optimization are close to the median values suggesting that the solution space is well behaved. Some large deviations in the 80th percentile curve occur primarily at higher extinction ratio values where extinction is also generally smaller and subject to higher measurement noise. Parameter value ranges are mostly fairly constrained for a given parameter, apart from the low extinction ratio bins for concentration ratio and the 1st mode median radius where there is a wider range of values. Generally, the second mode parameter values show significantly greater variance with many more outliers than the first mode. Some of the spread in these parameters may reflect geophysical processes, like volcanic events, so grouping data for analysis in ways that reflect the state of atmosphere may reduce the spread in the derived quantities. This will be pursued in the future.

The importance of the 2nd mode is a strong function of the extinction coefficient ratio with fraction of data in the 1st mode near 0.90 for the low altitude range and 0.8 for the high-altitude range for extinction coefficient ratios near 1 and with both increasing to over 0.95 for extinction coefficient ratios around 2 and essentially 1 for ratios above 4. The decrease in the importance of the second mode with increasing extinction coefficient ratio (and decreasing aerosol levels in general) is not surprising since the higher ratios indicate the scarcity of larger particles in the distribution. In these cases it is difficult for the WOPC measurements to separate the two modes as aerosol become increasingly small.

Figure 8 shows the values for SADR and VDR derived using the best fit parameters from the swarm optimization for both altitude ranges compared to the median and 20th and 80th percentile lines for SADR and VDR, as a function of 525 to 1020 nm extinction ratio, for all the WOPC data. The agreement between the median values for SADR and VDR with those from the swarm fits is reasonably good given the spread in the observed values and keeping in mind that they are multi-decade log plots. The range in SADR is larger in a relative sense than for VDR. This is almost exclusively due to a greater sensitivity of SAD to the variations in the number density of small particles compared to VD. As noted before, particles smaller than about 0.15 microns are ineffective scatterers and thus do not significantly impact aerosol extinction coefficient at SAGE II wavelengths.

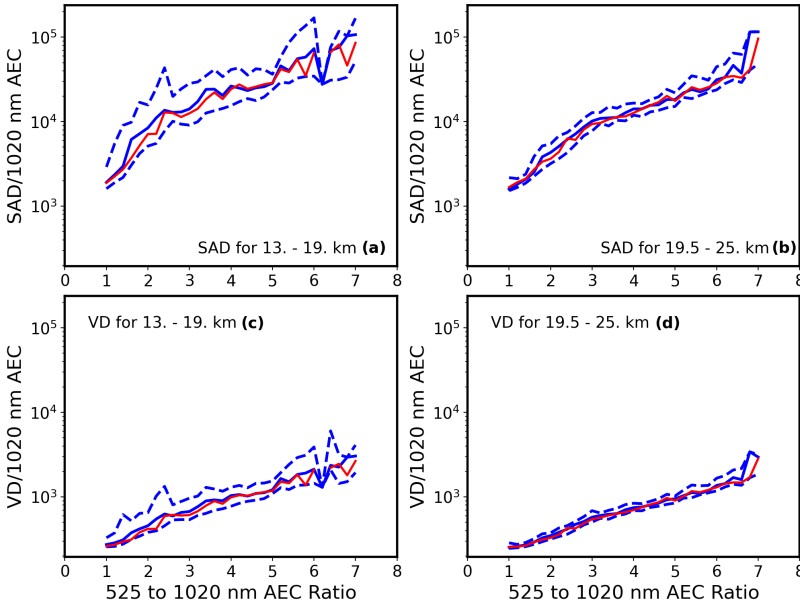

**Figure 8.** The ratios of SAD (a, b) and VD (c, d) to 1020 nm extinction coefficient as a function of the 525/1020 nm extinction ratio for 13-19 km (left) and 19.5-25 km (right) are shown. The continuous blue lines show the median with the dashed lines showing the and 20th and 80th percentile values for all WOPC data from Figure 4. The red line shows the dependence of the ratios of SAD and VD to 1020 nm extinction coefficient as a function of the 525 to 1020 nm aerosol extinction coefficient ratio for the swarm optimized fits.

Unsurprisingly then, it is clear that this approach does not capture the range of values shown in Figure 4 where the deviations from the median values can be very large. It is not overstating the case that it is impossible any approach based on SAGE-like measurements could infer such large variations. Generally, this outcome shows that the derived size distributions at least produce values for both parameters consistent with median WOPC values and that we have successfully derived a process by which a bimodal log-normal aerosol size distribution, consistent with the WOPC, can be assigned to all SAGE II observations. Assigning uncertainty to the swarm optimization technique would require further optimizations at each extinction ratio over the range of uncertainties in that ratio and is beyond the work undertaken here.

The strong dependence of SADR and VDR on extinction ratio shown in Figure 8 is in contrast to the relatively minor dependence of the lognormal parameters on extinction ratio shown in Figures 5, 6, 7. This is reflective of how sensitive SAD and VD are to small changes in median radii and distribution widths. Both SAD and VD have a highly non-linear dependence on median radii and distribution widths, which enter into the SAD and VD calculations through power law and exponential relationships. Thus seemingly small differences in median radii and width lead to large differences in SAD and VD.

A possible application for these derived size distributions could be in providing SAD and VD estimates as a product for the Global Space-based Stratospheric Aerosol Climatology (GloSSAC) (Thomason et al., 2018; Kovilakam et al, 2023). GloSSAC is a global, gap free aerosol climatology for the years 1979 through 2022. Data in this analysis is primarily from SAGE II but in the period immediately after the eruption of Mt. Pinatubo, some data is reconstructed using CLAES, HALOE, and ground-

330 based aerosol lidar data (Thomason et al., 2018). Aerosol extinction coefficients, at 525 and 1020 nm, are provided every half kilometer, every month, for latitudes centered at -77.5 to 77.5 in 5-degree increments.

Using the 525 and 1020 nm extinction and altitude of each data point, SAD and VD can be calculated from the corresponding SADR and VDR bin values by multiplying by the 1020 nm extinction value. Figure 9 shows those derived SAD and VD values calculated for 45° N (midlatitudes being the most applicable to WOPC size distributions) for the period around Pinatubo

through the quiescent period. Among the interesting features of these figures is the obvious abrupt increase of both SAD and VD below 19 km. This is expected based on the way we've approached the altitude dependence in this analysis and the significant differences we observe between the two altitude ranges. If one were implementing this approach as a retrieval algorithm, it would be beneficial to use more altitude groupings than the 2 used here.

For comparison Figure 10 shows SAD above Laramie, Wyoming, derived from various methods. In black, SAGE II v7.00

SAD along with it's uncertainty from Thomason et al. (2008). In blue, WOPC SAD values, median and 80th/20th percentiles, using only the OPC values discussed in this paper. In red, the swarm method derived values, median and 80th/20th percentiles. For SAGE II, the 525/1020 nm extinction ratio was calculated from measurements in a 5 degree latitude/longitude box around Laramie, Wyoming. For each SAGE II measurement in this region the swarm-derived PSD parameters associated with that extinction ratio were then used to find SADR (shown in Figure 8) and then multiplied by the SAGE II 1020 nm extinction

measurement to find SAD. Compared to the v7.00 SAD there is an improvement using the swarm method in the low altitude upper extinction ratio SAD's but a divergence for extinction ratio's < 3. At the upper altitudes both the swarm method and v7.00 SAD overestimate the WOPC SAD at extinction ratios > 5, otherwise the three estimates are quite similar. Investigating these differences is beyond the scope of this paper. The intention of this analysis has been to provide bimodal parameter values for SAGE II measurements, whereas Thomason et al. (2008) was to provide a reasonable range of SAD values for SAGE II

measurements with no realistic underlying particle size distribution.

## 6 Conclusions

Herein, we have used SAGE II and WOPC data to infer some of the limitations to inferring aerosol size distribution and some bulk properties solely from SAGE II and similar measurements. Based solely on WOPC measurements, we have inferred a median relationship between extinction ratio (525/1020) and the ratios of surface area and volume densities to 1020 nm extinc-

355 tion (SADR and VDR) that are broadly well-behaved, but that also exhibit substantial positive excursions that are effectively invisible to SAGE-like measurements, and thus cannot be reproduced in any quantitative way. We have derived representative bimodal log-normal size distribution parameters as a function of the 525/1020 extinction ratios using WOPC data. These data, in extinction ratio bins of width 0.2, were then used in a particle swarm optimization algorithm to generate bimodal size distribution parameters as a function of extinction ratio. The swarm derived distribution parameters and the inferred SADR and

360 VDR values are generally very close to the median values from the WOPC data. Overall, these bimodal size distributions may be useful in further applications, but care should be exercised since they are based solely on the behavior of data collected over Laramie, Wyoming, and may not be applicable at other latitudes. For instance, given the differences between the lower strato-

spheric values and the higher altitudes we observe herein, it is questionable in our minds how well these relationships would work in the lower tropical stratosphere where particle formation may be occurring. Additional issues may arise if sulfuric acid

aerosol is not the dominant aerosol type, such as following the Australian fires of 2019/2020 which was observed throughout the southern hemisphere stratosphere.

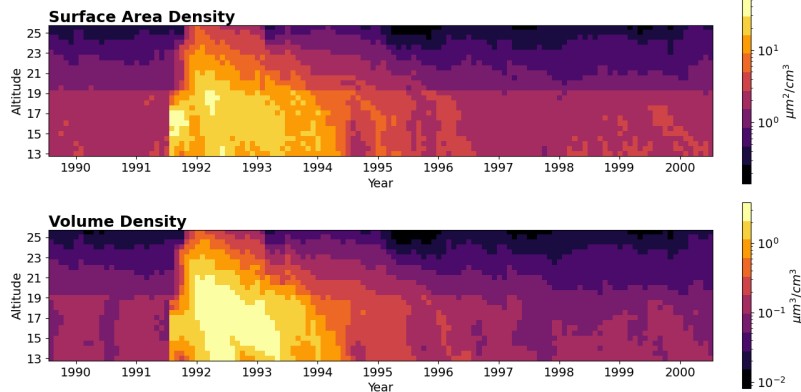

**Figure 9.** Surface area density (top) and volume density (bottom) at 45° N as a function of time, calculated using the GloSSAC 525 and 1020 nm extinction values and the corresponding SADR/VDR values calculated from the derived WOPC size distributions.

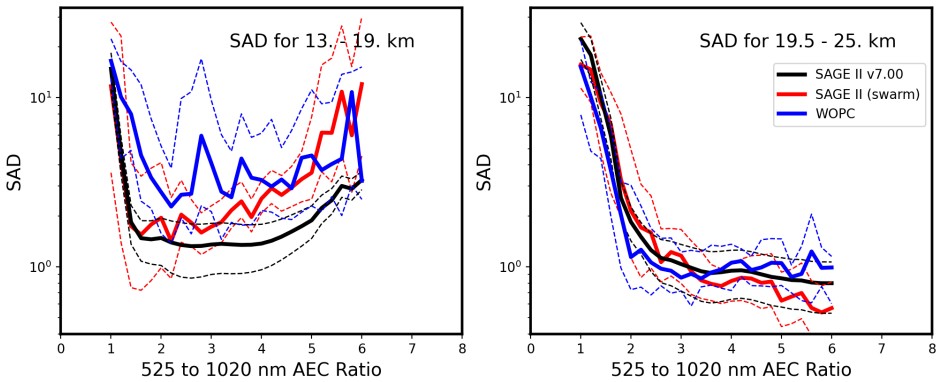

**Figure 10.** SAD median values, binned by AEC ratio, derived from three different methods. SAGE II v7.00 SAD (black) with it's reported uncertainty (black, dashed). WOPC reported SAD (blue) and the bin 80[th], and 20[th] percentiles (blue, dashed). The swarm method (red) uses the 525/1020 nm extinction ratio measured by SAGE II to find the SADR (seen in Figure 8) and then scaled by the measured 1020 nm extinction. Dashed red lines show the 80[th] and 20[th] percentile values for all SAGE II values calculated for a bin. For the swarm method and v7.00 SAD, measurements were used which were found in a +/- 5 degree latitude/longitude square centered on Laramie, Wy (41° N, 105° W).

*Code and data availability.* The code is available upon request to nicholas.a.ernest@nasa.gov. The data used in this paper is freely available at https://wyoscholar.uwyo.edu/collections/University_of_Wyoming_Stratospheric_Aerosol_Measurements/6379371 and https://asdc.larc.nasa.gov/project/GloSSAC for the UW OPC and NASA GloSSAC data sets respectively.

*Author contributions.* LT originated the concept for this research, provided direction for the analysis, and provided much of the commentary and interpretation of the results in this paper. NE conducted the analysis, developed the method for deriving mean size distributions, and provided some commentary. TD provided guidance on the use of WOPC data and commentary on the UW OPC project and the results of this paper.

*Competing interests.* The contact author has declared that none of the authors has any competing interests.

*Acknowledgements.* We would like to thank Travis Knepp for helpful comments made in producing this paper. TD acknowledges suport from SAGE III/ISS science grant.

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
