# Peer review of "Producing aerosol size distributions consistent with optical particle counters measurements using space-based measurements of aerosol extinction coefficient"

_Atmospheric Measurement Techniques, 2024_

## Author Response (AR1)

**Author's Response to Reviewer 1**

1. The section on information content seems somewhat underdeveloped. I appreciate the authors intent on clarifying why only two pieces of information are available, but this was shown in a more mathematical formulation by Thomason and Poole (1993). I would recommend clarifying what this analysis adds, or at least referencing that paper.

The new information in this paper is mostly an effort to demonstrate what was shown in the 1993 paper. People have ignored that paper for 3 decades and maybe it is time to reiterate that information without copying it. Referencing the 1993 paper is a good idea and the manuscript has been updated to include a reference to Thomason and Poole (1993). The following discussion of Thomason and Poole (1993) has been added to the paragraph that begins "If the ability of the 452 nm channel to illuminate variability in the ASD is low..": "As well a similar conclusion was reached by Thomason and Poole (1993) using a different technique."

2. As noted by the authors a limitation of the SAGE II data is the reliance on a single mode lognormal assumption to determine SAD/VD. However, the WOPC SAD/VD retrieval also makes a lognormal assumption. Is the insensitivity of SAGE to small particles the important consideration, or is the difference between a single vs bimodal fit the important distinction? See comment about Line 220 for more details. Line 220: While small scatterers do not directly contribute to SAGE measurements, if the lognormal assumption is correct, it seems they should be reflected in a change to the lognormal parameters. Is there a way to add small scatterers that changes the shape of the lognormal in a way that SAGE is insensitive to? Otherwise, if the small scatterers are present in a way that does not follow the lognormal distribution, what impact does this have on the WOPC retrievals, and the SAD/VD parameters derived here? (Also "poor scatters" -> "poor scatterers")

Our experience is that fitting a single mode log-normal size distribution to the SAGE II aerosol spectra often results in very narrow size distributions without an a priori constraint of what constitutes a satisfactory value for width. In fact, Thomason et al. (2008), demonstrated that these spectra can be fit reasonably well with a monodispersed (single radius) distribution. Realistically both single and bimodal distributions are approximations of the underlying size distribution. A bimodal distribution is more likely to capture the overall shape of the distribution than a single mode. Certainly, we find WOPC bimodal size distributions that produce extremely different values for SAD/extinction ratio for the same extinction ratio. The degree to which the log-normal assumption impacts WOPC fits is beyond the scope of this paper where our primary goal is 'to determine whether it is possible to infer the magnitudes and variability observed in WOPC-derived key parameters like SAD from SAGE II measurements.'

3. I don't understand the choice of a particle swarm algorithm to determine the best-fit parameters in Figure 5. I'm left wondering why the authors didn't take the mean counts in each WOPC bin and fit a bimodal distribution to that using the standard WOPC algorithm.

Using mean values for WOPC bins was actually the first method we used to derive characteristic aerosol size distributions for 525 to 1020 nm extinction bins. These efforts were ultimately not satisfactory as they did not produce size distributions parameters which reliably reproduced the extinction ratio of the bin or the parameter values of the distributions in the bin. This was an outcome of the complex interplay between the five parameters and the overlapping shapes of the size distributions. In many cases, the resulting size distribution not only did not reproduce the extinction ratio expected, it was almost disjointed with the input size distributions. Another method we tried used the mean parameter values as the representative parameter values for the bin, but this also did not reproduce the extinction ratio of the bin. Just like if you were to average the heights and widths of simple rectangles that all share the same area the derived rectangle using the mean height and width would not have the area they all have in common. There is no mathematical definition of what a mean shape should be, here we attempted to derive a mean shape for each bin in a way that also reproduced the bins extinction ratio. It is quite possible other solution methods would provide a satisfactory result but this approach worked well.

**Specific Comments**

Line 19: "...almost exclusively due to a broad range in particles below 0.15  $\mu$ m..." This cutoff of 0.15 $\mu$ m is not well substantiated in the paper. The WOPC have limited information below this value as well except for the condensation nuclei measurements (which are not always flown?).

This sentence has been changed to 'primarily due to variations in small radii particle number density. Roughly those smaller than about 0.15 um where the shortest wavelength extinction coefficient starts to drop off rapidly." There's nothing particularly special about the 0.15 micron value. These small particles with little effect on extinction at SAGE II wavelengths, in most contexts, have a significant effect on SAD. Say, if there are a particularly large number of small particles or if there is a relatively small amount above 0.15 microns.

Figure 2: Is the dotted line the relative or absolute uncertainty?

It is the relative uncertainty. The label has been changed to "452 nm median relative uncertainty" as this is more accurate.

Line 101: If I understand correctly, it is the variance between the predicted and measured 452nm signal that is important (as the absolute difference depends on the extrapolation model). However, I don't see this variance plotted in Figure 2, only the difference so I'm not sure how to interpret this figure.

The label has been updated to 'median relative aerosol extinction coefficient measurement uncertainty, (dotted, bottom scale)'.

The reviewer is correct that the standard deviation between the estimated and measured 452nm extinction coefficient gives the range of potential size distribution variations. However, all three lines are relevant to this discussion. The ratio of estimated to measured 453-nm extinction coefficient (dashed) is close to 1.0 below 23 km and is only 1.2 at 30 km. The standard deviation of the estimated and measured 452-nm extinction coefficient (solid) is always greater than the departure of the ratio from 1.0 and, most importantly, both are less than the measurement uncertainty (dotted). Our conclusion is that the 452-nm channel cannot add any additional information to any effort to infer the size distribution.

**Line 143: I think reference to Boone et al. (2023) is appropriate here.**

Reference to Boone (2023) has been included.

**Eq. 7: Should "particle value" be "parameter value"?**

Particle value is not quite right, it has been changed to "particle parameter value". Each particle has a parameter value associated with it (depending on where in the parameter space it is), this would be the value the particle has for a specific parameter.

Line 220: While small scatterers do not directly contribute to SAGE measurements, if the lognormal assumption is correct, it seems they should be reflected in a change to the lognormal parameters. Is there a way to add small scatterers that changes the shape of the lognormal in a way that SAGE is insensitive to? Otherwise, if the small scatterers are present in a way that does not follow the lognormal distribution, what impact does this have on the WOPC retrievals, and the SAD/VD parameters derived here? (Also "poor scatters" -> "poor scatterers")

While a better size distribution model would be really valuable, it is difficult to imagine that there would be sufficient information in the SAGE II data to account for a more complicated mathematical space from which to choose size distribution parameters. There is essentially one piece of 'size' information in the data. We find that it is impossible to account for all the variation we see in the WOPC. While we could use different models for the size distribution for the fits. At the end though, we'd expect the same shortcomings because the information to

craft a more robust answer just isn't there. The assumed size distribution, whatever it is, can only be fitted to the data available. Since small scatterers don't contribute to the SAGE II data there is no way to force any size distribution to be fit to the small particles below the instrument threshold. It only works for the WOPC due to the inclusion of the total aerosol concentration with the CN measurements.

**Line 236-237: Does neglecting unimodal conditions have an impact on potential SAGE II conversions? E.g. are unimodal fits more prevalent in background conditions biasing these SAD/VD conversions to more elevated aerosol levels, etc?**

We do not include OPC measurement sets where there are insufficient numbers of values to infer a bimodal distribution. The default approach of WOPC data processing solves for both a unimodal and bimodal fit, except when the large particle size bins do not have sufficient counts for the second mode retrieval. In the end the fit chosen to include in the data files is the one with the smallest RMS error. In most, but not all cases this is the bimodal fit. Unimodal fits occur mostly above the main aerosol layer. This does not mean that all fits using the bimodal model have two significant modes and, in fact, in a number of instances the second mode contributes little to number density distribution or bulk parameters like SAD. We don't believe this is an issue.

Figure 6: What parameters drive the SAD/VD dependence on extinction ratio? Perhaps it is just the plotting, but there seems to be little dependence on the lognormal parameters and mode fraction above extinction ratio values of 2-3 while the SAD/VD relationship remains clear.

This is a good question. The following text is added to the manuscript in section 5:

It should be noted that the strong dependence of SADR and VDR on R shown in Figure 8 is in contrast to the relatively minor dependence of the lognormal parameters on R shown in Figures 5, 6, 7 is reflective of how sensitive SAD and VD are to small changes in median radii and distribution widths. Both SAD and VD have a highly non-linear dependence on median radii and distribution widths, which enter into the SAD and VD calculations through power law and exponential relationships. Thus, seemingly small differences in median radii and width lead to large differences in SAD and VD.

**Author's Response to Reviewer 2**

The 2 modes presented in Figure 3 derived from the WOPC are shifted to smaller radii compared to the classical accumulation and coarse modes. A remark on and why that, and on the quantities listed in the legend, should be added. The values differ considerably from the swarm and median values in Figure 5. How many data points went into the optimization procedure?

The values in Figure 3 are based on measurements not a priori values. We have no expectation that they should fit classic accumulation and coarse mode radii, which are typically used to describe tropospheric aerosol size distributions. The sources and sinks of stratospheric aerosol do not produce distributions which fit readily into such a model. Figure 3 is just an example of a typical stratospheric aerosol size distribution within a few years of the Pinatubo eruption, so a remark on how such a distribution differs from a tropospheric distribution is unnecessary. The parameters in the legend are described in the new figure caption.

For figure 5, the number of WOPC distributions used in total for the lower altitude group was 1541. The number of distributions for the upper altitude group was 1515. The extinction ratio bins (for both altitudes) range in number from 25 to 105 distributions in a single bin.

To the paragraph discussing Figure 5 we add: "..., comprised of 1541 WOPC values for the lower altitude group and 1515 values for the upper altitude group."

**In the paragraph beginning in line 294 a cross-reference to Figure 4 with some text might be useful.**

The median, 80th, and 20th percentile values are the same values for Figure 8 as they are for Figure 4. Figure 8 has only cleared away the individual WOPC values and added the line which shows the SADR and VDR from the derived fits. The sentence in this paragraph has been changed from: "Unsurprisingly then, it is clear that this approach cannot capture the full variance in these parameters seen by the WOPC.", to "Unsurprisingly then, it is clear that this approach does not capture the range of values shown in Figure 4 where the deviations from the median values can be very large. It is not overstating the case that it is impossible any approach based on SAGE-like measurements could infer such large variations."

Also, the Figure 8 caption has been changed to the following: "Figure 8. The ratios of SAD (a, b) .... The continuous blue lines show the median with the dashed lines showing the and 20th and 80th percentile values for all WOPC data from Figure 4. The red line ..."

An additional frame in Figure 7 showing the surface area (SAD) provided in SAGE II data on NASA-EOSDIS in the same latitude region would demonstrate the improvements achieved by

**the new method. This can be also a line plot at some altitude (e.g. 18km) including the time series of the WOPC derived SAD and the one from SAGE II using old and new methods.**

Figure 10 has been added which shows SAGE II v7.00 SAD vs extinction ratio plotted against the values shown in Figure 8 scaled by the SAGE II 1020 nm extinction measurement. A description of this figure has been added to the paper.

**Specific comments**

**Line 37: Already here and maybe also in the abstract SAGE III should be mentioned since the presented approach is applied to GloSSAC later in the text.**

This seems appropriate, this work is being done now. This final line has been added to the Introduction section: "Work is ongoing now to extend this analysis to SAGE III/ISS."

**Fig.2 and lines 101ff: The discussed variance is not shown or is there a language problem?**

Text at lines 102 and 103 changed from 'variance' to 'relative RMS mean difference', and 'relative' added to line 110.

**Eqns. 3 and 4 both refer to channels. Why is there a different notation for 'N'? In Eqn.4 a 'j' is missing under the summation sign. It might be also not necessary to use 'a' instead of 'r' in the definite integrals.**

The j has been added to all three summations signs in these equations. Thank you. The a is just the integration variable defined by the limits on the integral and in Eqn. 3 it is used to distinguish the integration variable from r and  $r_{ch}$ . The a is maintained in Eqn. 4 for consistency.

In Eqn. 4 the notation on N has changed to  $N_{ch}$  because we can no longer claim  $N(r > r_{ch})$  since the integral is now from 0 to infinity, and CEFch(a) handles the extent to which  $r < r_{ch}$  and  $r > r_{ch}$  contribute to the integral. So this notation is maintained.

**Lines 241 and 279: Here 5 parameters are mentioned, in the abstract 6. Please add a clear remark why there is a difference.**

Clarification of the parameters has been added. The original line 241 has been changed to: "To simplify from 6 parameters, we scale all size distributions by the total particle number so that there are 5 free parameters to retrieve for bimodal distributions: ..."

**Figure 7: Why is the plot not to the end of the SAGE II observations where clear volcanic signals are visible?**

The plot is intended as an example and could span from 1984 through 2005 but at the expense of seeing some details in the Pinatubo recovery period from highly perturbed to quiescent.

**Line 306f: I suppose that here extinction data using SAGE II, CLAES, HALOE and ground-based instruments (Lidar) are used. It might be useful to include these details.**

The description has changed to: "GloSSAC is a global, gap free aerosol climatology for the years 1979 through 2022. Data in this analysis is primarily from SAGE II but in the period immediately after the eruption of Mt. Pinatubo, some data is reconstructed using CLAES, HALOE, and ground-based aerosol lidar data (Thomason et al., 2018). "'

**Technical corrections:**

**Line 87: ' $\lambda$ ' missing? Inconsistent to Figure.**

The y axis titles for Figure 1 have now been changed to  $Q\lambda(r,m\lambda)$  for graph (a) and  $3Q\lambda(r,m\lambda)/4r$  for graph (b).

**Line 94: What is the correct wavelength? Inconsistent to Figure.**

385 nm is correct. Figure 1 legend is changed.

**Line 221: Typo.**

Fixed.

**Eqn. 7: Typo?**

Fixed.

**Labels in figures often too small.**

Figure 3 axis labels enlarged. All text enlarged for Figure 4. Figure 5 has now been split into three figures to enlarge the graphs and clarify the text. Text has also been enlarged for Figure 8 as well as x and y-axis labels added.

**Figure 7: Please include the color steps in the figure in the color bar. Less steps would be better for identification of events.**

The color steps have been reduced to 10 to make the transitions clearer. Also, each step is now labeled next to the color bar.

**Lines 398, 400: Use upper and lower case for journal name.**

Fixed.

**Author's Response to Reviewer 3**

**1) Why is a particle swarm optimization algorithm used to infer ASD from R and not – what would be simpler – an empirical relationship between the median ASD and R?**

This method is attempting to define a median ASD based on bins of R. The median ASD parameter values for a bin often return extinction ratio values outside of the 0.2 ratio bin due to the non-linear relationship between parameter values and extinction ratio. While other methods for finding size distribution parameters would likely return similar results, we tried this one because it was interesting and it worked as it does both a good job deriving bimodal distribution parameters relative to the median WOPC measurement values, and it reproduces the target extinction ratio for a bin.

**2) There is only a marginal dependency between ASD and R. It is therefore questionable to derive a relationship from this. In contrast, the dependency between SAD/VD and R is strong. So, why is the ASD inferred from R and then used to calculate SAD and VD? Why are SAD and VD not derived directly from R?**

There is no direct route between extinction ratio and aerosol bulk properties without directly or indirectly making assumptions about the aerosol size distribution. We could simply correlate SAD and VD from the WOPC with computed R but even this indirectly is dependent on the size distributions inferred by the WOPC. In addition, we want the size distribution as an output because it provides egress to more aerosol attributes. The derived SADR/VDR based on the derived size distributions closely aligns with the WOPC values under most circumstances and shows the strength of the derived size distributions.

**3) Why is there a strong dependency between SAD/VD and R (Fig. 6), but a weak dependency between ASD and R (Fig. 5)?**

Because the relationship between SAD and VD and the size distribution parameters is particularly non-linear.

**4) On what basis was the intermediate altitude level (19 km-19.5km) chosen?**

A running mean of WOPC SAD values to extinction ratio informed the altitude groups. More than 2 altitude groups were also plotted to see how many altitude groups were needed, but significant overlap in values resulted in only 2 groups being chosen. WOPC values from 13 km were chosen as the minimum as this is generally near or below the tropopause. Size distribution fits to WOPC measurements are not provided below the tropopause. Values above 25 km are limited by low aerosol populations at larger sizes and thus generally unimodal distributions.

Specific comments:

**Line 5: "by the low information content" - please specify.**

Changed to ... by the restricted number of independent wavelengths of the SAGE...

**Line 37: Please indicate the name of the data record.**

This is not a single record, but was just referring to the general record of stratospheric aerosol encompassing many data sets including SAGE II. No change made.

Line 57: "While both modes do not necessarily contribute significantly to a computed aerosol extinction coefficient at SAGE II wavelength..."

Both modes contribute to the aerosol extinction coefficients. However, do the authors wanted to point out that the calculated aerosol extinction coefficient assuming a bimodal size distribution does not differ significantly from the aerosol extinction coefficient assuming a single size distribution?

This seems appropriate. The sentence has been changed from "While both modes do not necessarily contribute significantly to a computed aerosol extinction coefficient at SAGE II wavelengths, ..." to "Both modes do not necessarily contribute significantly to a computed aerosol extinction coefficient at SAGE II wavelengths, nor does the computed extinction at SAGE II wavelengths usually differ significantly between an assumed SLN distribution verses an assumed bimodal distribution."

**Line 61: Please specify "things".**

The line has been updated to: "...to infer aerosol bulk properties from SAGE II retrievals, this is not ..."

**Line 86: Please indicate the refractive index, temperature, and (?) water wapour content, at least in the figure caption.**

The figure caption was altered to include: " ...for the SAGE II channels assuming spherical water/sulfuric acid droplets at 220 K and a composition of 75% H2S04 and 25% H2O. The real refractive indices used were 1.432, 1.432, and 1.421 for 453, 525, and 1020 nm respectively with zero for all imaginary parts."

Line 109, Figure 1: Was only one Angstrom coefficient calculated and used to calculate all extinction coefficients at 452 nm? Or was individual Angstrom coefficients determined from each measurement and used to calculate the extinction coefficients? If the latter, why does the Angstrom extrapolation work better in some altitude ranges than in others? Can reasons be given for this?

A separate Angstrom coefficient was computed for each set of observations at 452, 525, and 1020 nm. Using the Angstrom coefficient to represent the wavelength dependence of aerosol extinction coefficient is a pretty simple model and, while it approximately captures observed behavior, it misses a curvature observed in measured spectra and spectra modelled using the usual assumption of single mode log-normal size distributions. One could speculate on the altitude dependence. The dependence appears above 25 km where there are fewer large particles so the 1020 nm extinction coefficient would be low. But such an investigation is not the focus of this plot, rather it was to demonstrate the dependence of the 452 nm channel on the upper channels thereby limiting its usefulness as an independent measurement.

**Line 130: limb measurements $\rightarrow$ occultation measurements?**

In this case we are referring to limb-scatter measurements (e.g. OMPS and OSIRIS). Clarified in the text, this line has been changed to: "While we are focused on SAGE II, it is worth considering whether other measurement types such as the limb-scatter technique employed by OSIRIS can be used to infer aerosol size distributions in a similar way."

**Equ. 3, 4: Description of parameter "a" is missing.**

"a" is the integration variable which spans the limits on the integrals indicated by the subscripts and super scripts. To eliminate confusion the following text is added just before Eqn 3. ... size distribution is given by the following integrals where a is the integration variable.

**Equ. 5: "r" should be "a"?**

See above.

**Fig. 3: Numbers and parameters in the legend need appropriate descriptions.**

Descriptions of parameters have been added to the caption.

**Line 216: "infer SAD from SAGE II measurements". Should rather be "infer SAD from R" because "SAGE II measurements" could mislead the reader since Fig. 4 shows WOPC data.**

This was intended to be a general statement about SAGE II measurements, it wouldn't be the same message to say "infer SAD from R".

**Line 229: "under some conditions total volume estimates can be inferred". Which conditions?**

This phrase is deleted since it is not relevant. The condition is that if we have sufficient constraint from in situ observations which is the rest of the sentence.

Line 243: Can the authors briefly describe what a particle swarm optimization algorithm does in physical terms?

A brief description was added to the sentence "To retrieve these parameters, we employ a particle swarm optimization algorithm (Hu and Eberhart, 2002), where the many individual sets of WOPC data are referred to as particles in the algorithm's nomenclature, and the particles "fly" through parameter-space trying to minimize or maximize some function". The following paragraphs should serve better to clarify this.

**Equ. 6, Line 249:**

**• Are r\_..., s\_..., f\_err, and R\*w absolute, i.e., positive values?**

They are absolute values. Equation 7 has been updated to include the absolute value notation. As well this line has been added just before Equation 7: "The error values are all positive as they are based on the absolute value of the difference between the particle's parameter value and the parameter median, divided by the parameter standard deviation, defined here:".

**• I think r, s, f, and R are not the two mode radii, widths of modes, ratio, and the center of an extinction ratio bin, respectively, but the ERRORS of these values.**

To avoid confusion the description of the variables has been changed to : "where for the two modes r\_01\_err/r\_02\_err are the median radii errors, s\_01\_err/s\_02\_err are the width errors, f\_err is the error for the ratio of the concentration of the first mode to the total concentration, R\_err is the extinction coefficient ratio error, and w is a weight.". This now explicitly references the variables of the equation.

**r, s, and f have different values depending on the numerical range of the corresponding parameters. Shouldn't a weighting factor be built in that weights the parameter errors accordingly? Can they be weighted equally? Or should errors in r, for example, be weighted more heavily than errors in s?**

They are weighted by the standard deviation, this is shown in the following parameter error equation (7).

**Line 278: Typo: Figure 5**

This has been fixed. (now Figure's 5, 6, 7)

Line 299: "SAGE measurement". Should be WOPC measurement, since only WOPC data are shown. - Or do the red lines in Figs. 5 and 6 show the results using SAGE II aerosol extinction ratios?

The blue (dotted and continuous) lines of Figure 6 (now Figure 8) represent WOPC values sorted by SAGE extinction ratios. This range of values, represented in the dotted lines, is caused by lack of information in a SAGE measurement which is what "SAGE measurement" is referencing. So "SAGE measurement" is correct.